

# Viable bacterial communities on hospital window components in patient rooms

Patrick F. Horve[1], Leslie G. Dietz[1], Suzanne L. Ishaq[1,2], Jeff Kline[1], Mark Fretz[3] and Kevin G. Van Den Wymelenberg[1,3]

[1] Biology and the Built Environment Center, University of Oregon, Eugene, OR, United States of America
[2] School of Food and Agriculture, University of Maine, Orono, ME, United States of America
[3] Institute for Health in the Built Environment, University of Oregon, Portland, OR, United States of America

Corresponding authors
Patrick F. Horve, pfh@uoregon.edu
Kevin G. Van Den Wymelenberg, kevinvdw@uoregon.edu

## ABSTRACT

Previous studies demonstrate an exchange of bacteria between hospital room surfaces and patients, and a reduction in survival of microorganisms in dust inside buildings from sunlight exposure. While the transmission of microorganisms between humans and their local environment is a continuous exchange which generally does not raise cause for alarm, in a hospital setting with immunocompromised patients, these building-source microbial reservoirs may pose a risk. Window glass is often neglected during hospital disinfection protocols, and the microbial communities found there have not previously been examined. This pilot study examined whether living bacterial communities, and specifically the pathogens Methicillin-resistant *Staphylococcus aureus* (MRSA) and *Clostridioides difficile (C. difficile),* were present on window components of exterior-facing windows inside patient rooms, and whether relative light exposure (direct or indirect) was associated with changes in bacterial communities on those hospital surfaces. Environmental samples were collected from 30 patient rooms in a single ward at Oregon Health & Science University (OHSU) in Portland, Oregon, USA. Sampling locations within each room included the window glass surface, both sides of the window curtain, two surfaces of the window frame, and the air return grille. Viable bacterial abundances were quantified using qPCR, and community composition was assessed using Illumina MiSeq sequencing of the 16S rRNA gene V3/V4 region. Viable bacteria occupied all sampled locations, but was not associated with a specific hospital surface or relative sunlight exposure. Bacterial communities were similar between window glass and the rest of the room, but had significantly lower Shannon Diversity, theorized to be related to low nutrient density and resistance to bacterial attachment of glass compared to other surface materials. Rooms with windows that were facing west demonstrated a higher abundance of viable bacteria than those facing other directions, potentially because at the time of sampling (morning) west-facing rooms had not yet been exposed to sunlight that day. Viable *C. difficile* was not detected and viable MRSA was detected at very low abundance. Bacterial abundance was negatively correlated with distance from the central staff area containing the break room and nursing station. In the present study, it can be assumed that there is more human traffic in the center of the ward, and is likely responsible for the observed gradient of total abundance in rooms along the ward, as healthcare staff both deposit more bacteria during activities and affect microbial transit indoors. Overall, hospital window components possess similar microbial communities to other previously identified room locations known to act as reservoirs for microbial agents of hospital-associated infections.

# BACKGROUND

Hospital-associated infections (HAIs) are a leading cause of hospital patient morbidity and mortality in the United States (*Murphy, Whiting & Hollenbeak, 2007*; *James, 2013*). The five most common hospital-associated infections include: central line-associated blood infections, ventilator-associated pneumonia, surgical site infections, *Clostridioides difficile* infections (CDI), and catheter-associated urinary tract infections. These conditions cost the U.S. healthcare system approximately $3.3 billion annually (*Zimlichman et al., 2013*) and extend hospital stays by approximately two-fold for patients (*Glance et al., 2011*). With HAIs increasing overall costs and placing a burden on both the healthcare industry and patients, a further understanding of HAI causes and interventions is warranted. Notably, only ∼50% of HAIs can be attributed to a specific source (*Magill et al., 2014*). Given that many HAIs are caused by opportunistic pathogens which take advantage of host immune system alterations and disruption of homeostasis, a better understanding of microbial reservoirs in proximity to the patient and transmission dynamics within the hospital may yield a more effective strategy towards reducing incidences.

Microbial exchange occurs between hospitals and patients. Patients seed hospital rooms with their unique microbial communities (*Best et al., 2010*; *Horve et al., 2019*), and are exposed to and colonized by microorganisms from the hospital room itself (*Prussin 2nd, Garcia & Marr, 2015*; *Jou et al., 2015*; *Mcdonald et al., 2016*; *Lax et al., 2017*). Microorganisms deposited from both current and previous occupants can be aerosolized from surfaces by the daily activities of patients, visitors, and healthcare staff (*Rashid et al., 2017*) and come into direct contact with occupants, or may resettle on surfaces and among dust where they may survive for months (*Haysom & Sharp, 2003*). While the transmission of microorganisms between humans and their local environment is a continuous exchange which generally does not raise cause for alarm, in a hospital setting with immunocompromised patients, these building-source microbial reservoirs may pose a risk.

Aerosolization of microorganisms can act as a transmission vector for HAIs (*Pierce & Sanford, 1973*). Pathogens have been found to be aerosolized from walking (*Qian, Peccia & Ferro, 2014*), toilet use (*Best, Sandoe & Wilcox, 2012*; *Aithinne et al., 2019*), and cleaning activities (*Best et al., 2010*), sending microorganisms throughout the room. Although patient rooms are cleaned following a set protocol, any additional cleaning protocols that are enacted are only done so in response to the patient's diagnosis. If a patient is not diagnosed with a containment-risk pathogen, only the standard cleaning protocols are followed. It has been demonstrated that despite cleaning efforts, pathogens can persist in hospital rooms at concentrations sufficient for transmission (*Boyce et al., 2009*; *Otter, Yezli & French, 2014*). Furthermore, compliance with cleaning protocols has the potential to impact transmission (*Alfa et al., 2008*; *Horve et al., 2019*). Often, windows and associated

components are absent from hospital disinfection protocols (*Mehtar, Hopman & Duse, 2018*). Due to the potential for microbial dispersal, an understanding of microbial dust communities of all locations within a hospital room is essential.

There has been no characterization of the microbial communities of window glass and associated components within hospital rooms. Our previous study demonstrated that light negatively impacts the survival of microorganisms in dust within built environments (*Fahimipour et al., 2018*), and it is postulated that daylighting could be used to control microbial populations indoors. To better understand the relative risk to patients from patient room windows and associated components, this study sought to identify (1) whether viable (intact) microorganisms were present on window glass, window frame surfaces, and room curtains; (2) whether relative light exposure (direct sunlight or not) was associated with differences in the viable microbial communities on hospital surfaces; and (3) if light exposure would alter the abundance of two high-risk, hospital-associated pathogens; *Clostridioides difficile (C. difficile)* and methicillin-resistant *Staphylococcus aureus* (MRSA).

## MATERIAL AND METHODS

### Sample collection

University of Oregon laboratory personnel trained Oregon Health & Science University (OHSU) medical staff on performing sampling protocols, to reduce interference in patient rooms and protect patient privacy. Environmental sampling does not require Institutional Review Board approval; however, written information on the project including what samples were being collected, how they would be used, and if the samples would impact patient privacy, was disseminated by OHSU medical staff and discussed with patients prior to sampling.

Samples were collected between 10:00 a.m. and 11:00 a.m. on June 7, 2019, from the 13th floor of Kohler Pavilion (13K) at OHSU in Portland, Oregon, USA (Fig. 1A). All patient rooms have a window with anodized aluminum sill, jamb, and head frame components, and double-pane insulated glazing units (Fig. 1B). Each window includes a vinyl-backed, hanging curtain on a track which extends beyond the window along the entire outer wall, allowing the curtain to be pulled completely away from the window along the wall. Curtains are laundered after a room is occupied by a patient flagged for a biohazardous infection, such as CDI. Upon patient discharge, OHSU Environmental Services follow biohazardous infection cleaning protocols. Sampling locations within patient rooms included window glass surface (presumed direct sunlight), window-side of the curtain (presumed direct sunlight), patient-side of the curtain (presumed indirect sunlight), glazing-side of the window frame at the sill (presumed direct sunlight), the window frame surface facing into the room at the sill (presumed indirect sunlight), and wood-covered air return grille (Fig. 1B).

Environmental samples were collected using Copan FLOQSwabs (Copan Diagnostics, Catalog #519CS01) pre-moistened with sterile 1X phosphate buffered saline (PBS) solution. The sampled area was swabbed using an overlapping, horizontal "S" pattern and a single-use, disposable plastic frame was used to standardize the collection area. Swabbed areas

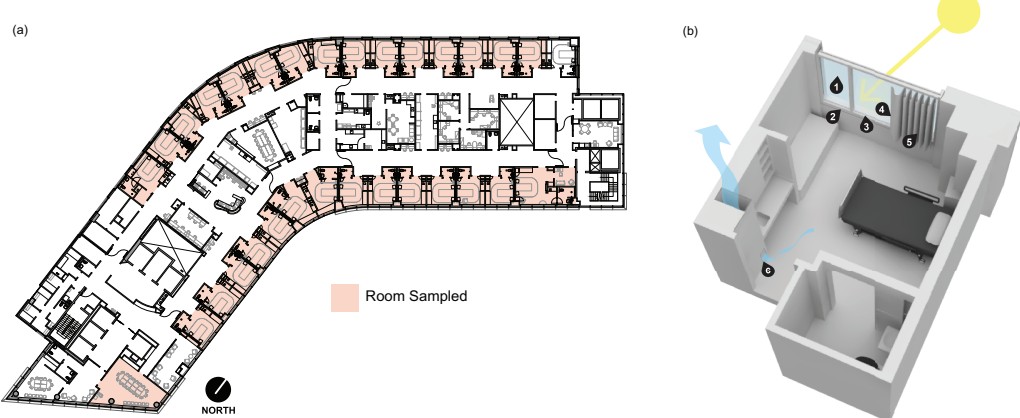

**Figure 1  Floor plan and rendering of a typical patient room at the Oregon Health and Science University hospital.** (A) Floor plan of the 13th floor of Kohler Pavilion (13K) at Oregon Health and Science University (OHSU). Red shading indicates the rooms that were sampled between 10:00 a.m. and 11:00 a.m. on June 7, 2019 (b) Digital rendering of a typical patient room on OHSU (13K) with the sampling locations indicated by the numbers. The sampled locations were (1) window glass surface, (2) the window frame surface facing into the room at the sill, (3) glazing-side of the window frame at the sill, (4) window-side of the curtain, (5) patient-side of the curtain and, (6) wood-covered air return grille.

were located 152 cm to 183 cm off the ground (average person height), with the exception of the air return grilles which were located 30 cm to 45 cm off the floor. A total of 182 samples were taken from 29 patient rooms and one conference room. Five control samples were collected concurrently, comprised of two PBS controls (swab soaked with sterile 1X PBS), an air control (soaked with sterile 1X PBS and waved in the air for approximately 30 s), and two negative controls swabbed from sampling staff nitrile gloves to account for staff-sourced contamination, resulting in 187 samples total. After collection, swabs were sealed in individual sterile transport tubes with a 2 mL aliquot of sterile 1X PBS each and immediately placed into an ice-filled transport cooler.

## Genomic material preparation

Following collection, all samples were maintained at 4 °C and processed in randomized batches within 48 h. Swab tips and 1 mL of 1X PBS from the transport tube were placed into a sterile 2 mL Eppendorf tube and vortexed briefly to resuspend as much dust as possible from the swab tip. The swab tip was then removed from the tube, leaving the 1X PBS and resuspended dust particles behind. In order to quantify viable cells only, samples were treated with propidium monoazide (PMA) (*Nocker et al., 2007*). PMA infiltrates dead or damaged cells through disrupted cell walls and membranes and binds to DNA, preventing polymerase-chain reaction (PCR). One 10 μL aliquot of 2.5 mM PMA solution (Fisher Scientific, Catalog #NC9734120) was added to the 1X PBS and dust solution (final concentration of 25 μM), incubated in the dark for 10 min, vortexed briefly, and incubated in blue light using a PMA-Lite LED Photolysis Device (OPE Biotechnology

Co., Ltd., Model #PT-H18A) according to the manufacturer protocol. DNA extractions were performed using DNeasy PowerLyzer PowerSoil Kit (Qiagen, catalog #12855-100), following manufacturer protocol.

## Molecular analysis

MRSA abundance was quantified using quantitative PCR (qPCR), with DNA standards (Integrated DNA Technologies) comprising a 200 bp segment of the MRSA nuc gene with SANuc F primer (5′-TAAAGCGATTGATGGTGATACG-3′) and SANuc R primer (5′- TTCTTTGACCTTTGTCAAACTCG-3′) (*Bamigboye, Olowe & Taiwo, 2018*). Thermocycling conditions were as follows: 95 °C for 5 min, 40 cycles of 95 °C for 15 s , 60 °C for 50 s , and 72 °C for 30 s. *Clostridioides difficile* abundance was quantified using qPCR with DNA standards from Integrated DNA Technologies (Coralville, Iowa, USA) comprising a 186 bp segment of the TcdA gene with TcdA F primer (5′- CAGGACACACAGTGACTGGTAA-3′) and TcdA R primer (5′- GAACTGCTCCAGTTTCCCAC-3′) that were designed using the National Center for Biomedical Information (NCBI) Primer-Blast tool and the *C. difficile* TcdA gene. Thermocycling conditions were as follows: 95 °C for 5 min, 40 cycles of 95 °C for 15 s, 60 °C for 50 s, and 72 °C for 30 s. Total bacterial abundance was quantified using DNA standards (Integrated DNA Technologies) comprising a 167 bp segment of the 16S rRNA gene using Total Bacteria F SYBR Primer (5′- GTGSTGCAYGGYTGTCGTCA-3′) and Total Bacteria R SYBR Primer (5′-ACGTCRTCCMCACCTTCCTC - 3′) (*Fahimipour et al., 2018*). DNA sequences for standards are provided in Table S1. Thermocycling conditions were as follows: 95 °C for 5 min, 40 cycles of 95 °C for 15 s, 58 °C for 50 s, and 72 °C for 30 s. All qPCR plates were prepared using an Eppendorf epMotion 5075 robot. PowerUp SYBR Green Mastermix (Thermo-Fisher Scientific, Catalog #A25741) and an ABI QuantStudio3 (Applied Biosystems, Catalog #A28137) were used to detect amplification of targeted gene regions. Standard curves were generated using serial-dilutions of the synthetic DNA standards of the SANuc gene ($R^2 = 1$), TcdA gene ($R^2 = 0.98$), and 16S rRNA gene ($R^2 = 1$) with known gene sequence copy numbers.

Genomic DNA was amplified for high-throughput DNA sequencing, targeting the V3/V4 hypervariable region of the 16s rRNA gene, using the primers pair 319F/806R (*Drewes et al., 2017*), dual-indexed barcode primers (Integrated DNA Technologies), and NEBNext High Fidelity 2X Mastermix (New England Biolabs, Catalog #M0541). Thermocycling conditions were as follows: 98 °C for 5 min, 40 cycles of 98 °C for 15 s, 60 °C for 50 s , 72 °C for 30 s , with a final elongation of 72 °C for 2 min. Primer and PCR reagents were removed using Mag-Bind RxnPure Plus beads (Omega Bio-Tek, Catalog #M1386). The resulting amplicons were quantified using Quant-iT dsDNA High Sensitivity Assay Kit (Thermo-Fisher Scientific, Catalog #Q33120) on a Molecular Devices SpectraMax M5E Microplate Reader and pooled to 40 ng DNA per sample, then sequenced at the University of Oregon's Genomics and Cell Characterization Core Facility (GC3F). Sequencing was performed on an Illumina Miseq using V3 chemistry, generating $2 \times 300$ nt reads.

## Statistical analyses

Raw Illumina sequence data were filtered, trimmed, and denoised using the *DADA2* v1.8.0 statistical inference algorithm (*Callahan et al., 2016b*; *Callahan et al., 2016a*) using the R platform (*R Core Team, 2020*), which identifies ribosomal sequence variants (RSVs). Due to poor sequencing quality, forward reads only were used and were truncated to 125 nt, required to have no ambiguous bases, and each read was required to have fewer than two expected errors based on quality scores. Taxonomy was assigned to RSVs using the RDP Bayesian classifier implemented in *DADA2* against the Silva (*Quast et al., 2013*) version 132 reference database. This method is shown to have an accuracy of 77.8% confidence for sequence queries of 125 nt for identifying taxonomy at the genus level (*Wang et al., 2007*). Prior to analyses, we removed variants classified as mitochondria or chloroplasts.

To remove putative contaminants, we utilized the *decontam* (*Davis et al., 2018*) R package, which utilizes statistical inference and sequenced negative controls to identify putative contaminating sequence features from high-throughput sequencing data. The effects of lighting conditions and sampling locations on community compositions were quantified using a permutational multivariate analysis of variance (PERMANOVA) with the *vegan* package (*Oksanen, 2018*) using variance stabilized reads. Pairwise contrasts between treatment groups were accomplished by performing PERMANOVA analyses with 10,000 matrix permutations for each pair of factor levels. Differential microbial abundance between sampling locations and lighting conditions was performed using the DESeq2 R package on non-rarefied reads (*Love, Huber & Anders, 2014*). DESeq2 utilizes shrinkage estimations for dispersions and fold changes for differential analysis of count data, essentially determining when different bacteria are present in significantly different amounts. Significant differences in the distribution of alpha diversity values between each sampling location were tested using two-sample Kolmogorov–Smirnov tests.

## Data and code availability

The raw sequencing data from this project have been uploaded to the National Center for Biotechnology Information (NCBI) Sequence Read Archive (SRA) and are freely accesible under BioProject ID PRJNA610453. The code for all analyses and raw abundance data is available at https://github.com/BioBE/Windows-as-Potential-HAI-Reservoirs.

# RESULTS

## Abundance

Viable bacteria were found to occupy all locations sampled. There was no significant difference ($F_{3,66} = 0.811$, $p = 0.49$) in viable bacterial absolute abundance between sampling locations within the hospital rooms (Fig. 1A). Across all samples, there was also not a significant difference ($F_{1,68} = 0.824$, $p = 0.36$) in abundance of viable bacteria between surfaces exposed to direct sunlight and those which were not exposed to direct sunlight (Fig. 2B). However, sunlight exposure indoors is strongly mediated by window orientation, thus the analysis was further disaggregated by cardinal orientation. Combining samples from all surfaces, rooms with windows that were facing west demonstrated a higher abundance of viable bacteria than those facing northwest ($p = 0.00039$), east ($p = 0.035$),

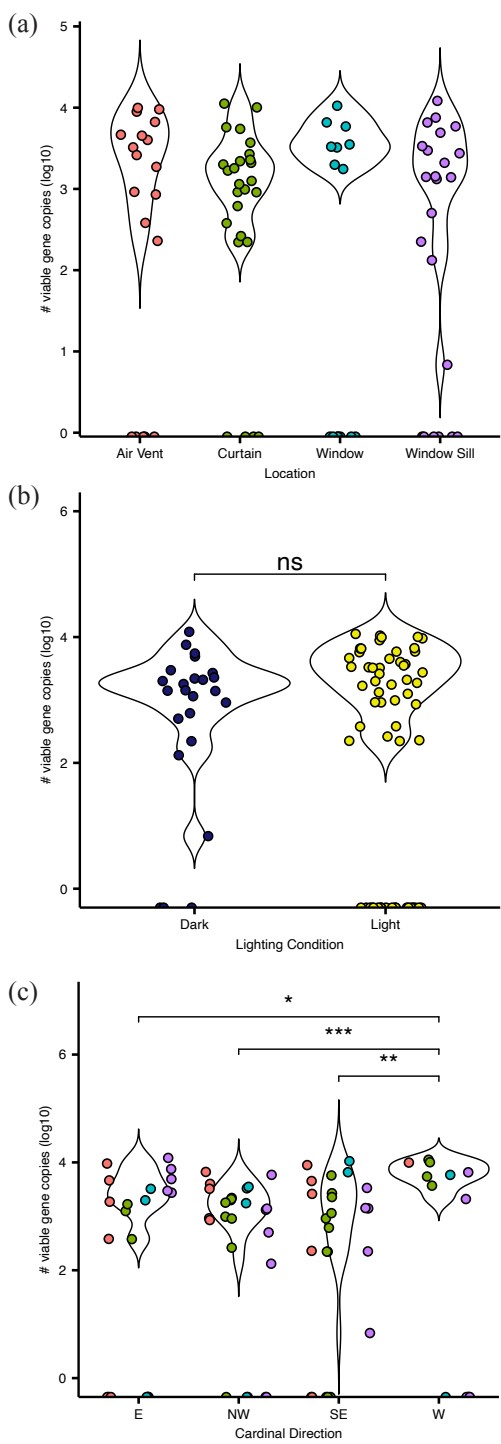

**Figure 2** **Absolute abundance of bacteria at sampling locations around windows within hospital patient rooms.** Each point represents an individual sample, calculated as qPCR-based estimates of $\log_{10}$-transformed copies of the 16S rRNA gene. Abundance shown by (A) room sampling locations, including air vents (red), curtains (green), windows (turquoise), and window sills (purple); by (B) relative light exposure, including surfaces in indirect sunlight (navy blue) and direct sunlight (yellow); and by (C) window orientation of cardinal direction, colors are the same as in (A).

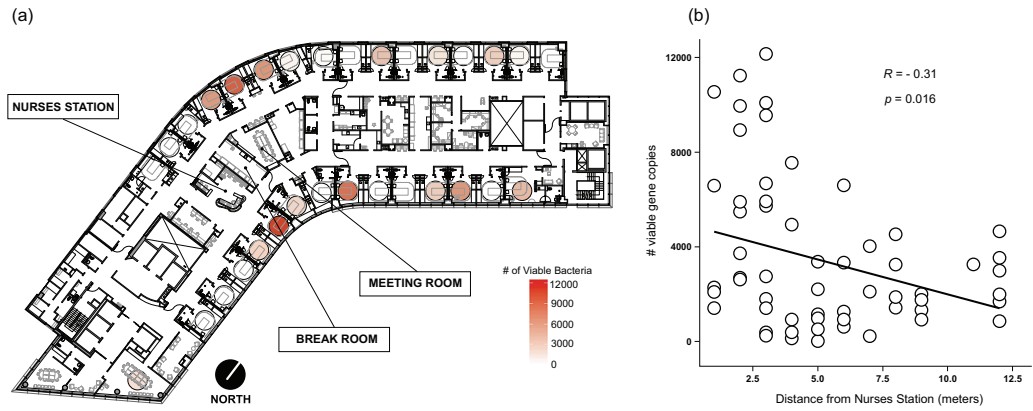

**Figure 3 Absolute abundance of bacteria around windows by room across a hospital ward.** Absolute bacterial abundance was calculated with qPCR-based estimates of log $_{10}$-transformed copies of the 16S rRNA gene. Abundance at all sampled locations within the patient room were summed to obtain total bacteria abundance. Abundance is shown (A) overlaid on a floor plan of ward 13K at OHSU and (B) as a function of the distance of each room from the centrally-located nurse station on their half of the floor.

and southeast ($p = 0.0065$) (Fig. 2C). Summary statistics are provided in Table S2. Rooms with the highest bacterial abundance were located directly next to the nurses station, break rooms, and conference rooms (Fig. 3), while rooms located at the ends of hallways demonstrated lower viable bacterial abundance ($p = 0.016$). Viable *C. difficile* was not detected at any of the sampled locations. Viable MRSA was detected at very low abundance in 2.4% (2/83) of the samples collected from the curtain in direct sunlight (∼3 viable gene copies) and the window sill in direct sunlight (∼17 viable gene copies); there was no detectable live MRSA on the glass, window frame, or air return grille.

## Community composition

Overall, there was no significant difference ($p = 0.083$) in alpha diversity between locations in direct sunlight and those not in direct sunlight (Fig. 4A), as measured by the Shannon Index (Shannon H′), which incorporates both the richness and evenness of the community. There was no significant difference in diversity at specific locations with presumed differential amounts of sunlight. However, the bacterial communities present on the surface of the glass were significantly ($p = 0.03$) less diverse than other communities within the patient room (Fig. 4B). There was no significant difference in the observed distribution in diversity values between the windows and the curtain, but a significant difference was observed between the window and air return grille (ks, $p = 0.048$) and the window and the window sill (ks, $p = 0.023$), indicating different distributions in alpha diversity by sampling location (Fig. 4B). There were no significant differences in the diversity of the microbial community between the air return grille, the window sill, or either side of the curtain.

A total of 1,023 unique RSVs were identified across all environmental swabs. The most relatively abundant phyla based on variance stabilized reads were Proteobacteria, Actinobacteria, and Firmicutes, with no major differences at the phylum level based

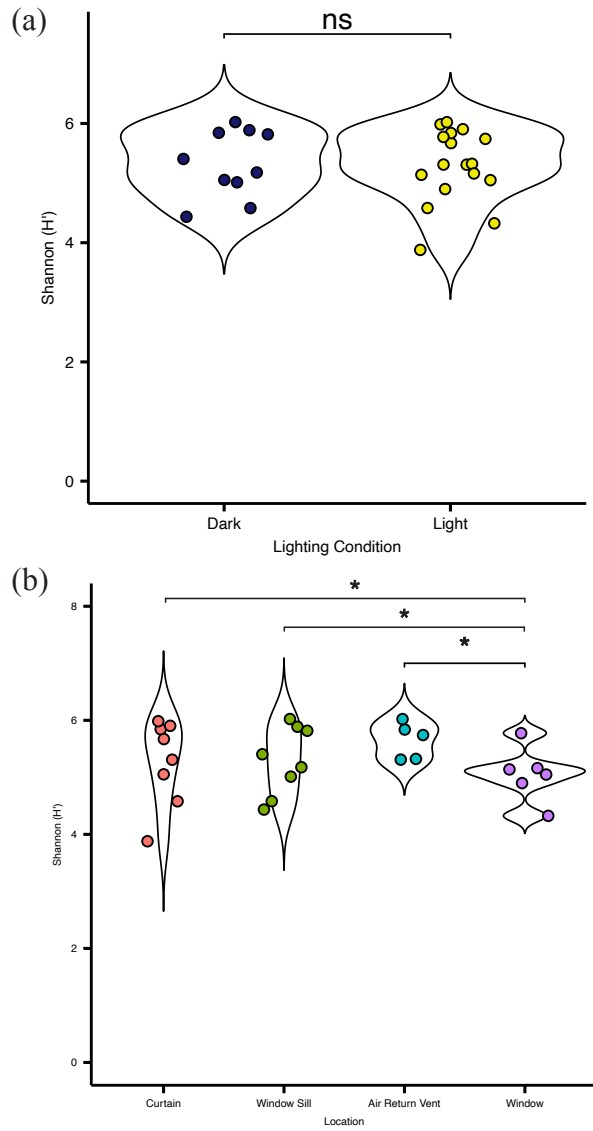

**Figure 4** **Alpha diversity of viable bacterial communities from locations around hospital room windows.** Shannon diversity (H') from (A) surfaces in indirect sunlight (navy blue) and direct sunlight (yellow); and (B) from room sampling locations, including curtains (red), window sills (green), air return vents (turquoise), and windows (purple).

on location within the patient room or presumptive amount of direct sunlight received (Fig. S1C).

At the genus level, all locations (Fig. S1D) and lighting conditions (Fig. S1D) were very similar in their overall microbial composition. The *Corynebacterium*, *Methylobacterium*, *Pseudomonas*, and *Nocardioides* genera were most abundantly represented across all locations and lighting conditions within patient rooms. There were 47 individual genera that were differentially abundant ($p < 0.05$) across all samples. The air return grille demonstrated a large variety of bacteria that were differentially more abundant
(Figs. 5A–5C). The surface of the glass also demonstrated a large variety of differentially abundant bacteria. *Acidaminococcus, Bilophila, Curtobacterium, Eremococcus, Luteilobacter, Megasphaera, Polymorphobacter, Rheinheimera, Rhodocytophaga, Rothia, Staphylococcus, Stenotrophomonas*, and *Veillonella* were all differentially abundant on the glass compared to the air return grille (Fig. 5C), curtain (Fig. 5D), and window sill (Fig. 5E). There were very few differentially abundant bacteria between locations in direct sunlight and those not in direct sunlight, including increased observed abundance of *Bosea* and *Nesterenkonia* in indirect sunlight and decreased observed abundance of *Pseudomonas* and *Veillonella* in direct sunlight (Fig. 5F). The relative abundance of discriminant taxa at each location around the windows are shown in Fig. 5G.

## DISCUSSION

To our knowledge, this is the first study which characterizes the microbial communities of hospital windows and components, and acts as a first step toward a better understanding of the relative risk that patient room windows and associated components pose to patients. Using quantitative and qualitative microbial assessment, this study identified (1) viable (intact) microorganisms present on window glass, window frame surfaces, and room curtains; (2) that relative light exposure (direct sunlight vs. indirect sunlight) was associated with changes to the viable microbial communities on hospital surfaces; and that (3) sunlight exposure did not alter the abundance of two high-risk, hospital-associated pathogens, *C. difficile* and MRSA. However, cardinal direction of the window did have an effect, which implies that there is a time or intensity threshold to sunlight exposure for altering indoor microbial communities. Further, this study suggests a spatial pattern to bacterial abundance based on putative occupancy density, potentially implying that human occupants may have a stronger effect on indoor microbial communities than ambient light exposure.

Overall, viable bacterial load was similar across all locations, but lower bacterial diversity was found on the surface of the window glass. We hypothesize nutrient available or surface adhesion might play a role in the reduced richness observed on the glass, but these were not evaluated in the present study. The more abundant bacterial taxa on windows identified in this study are associated with unique, advantageous means of energy production. For example, *Acidaminococcus* is able to utilize amino acids as an energy source (*Rogosa, 1969*) and *Polymorphobacter* is capable of photosynthetic energy production (*Hirose, Matsuura & Haruta, 2016*). A previous study of bacterial communities in dust under different lighting conditions suggest that some bacteria can survive in enclosed systems, either as auxotrophs or by cannibalizing other members of the microbial community. The present study did not examine bacterial survival over time, or bacterial metabolism, but this hypothesis represents a clear next step in hospital microbiome research.

Notwithstanding the differentially abundant bacteria between each of the room surfaces, the overall composition of the bacterial communities are very similar, with very few outlier taxa among the 25 most abundant bacteria, regardless of location or light treatment. Even when comparing these 25 bacteria in each sample that are also considered differentially abundant, very few differences were observed between location and lighting. Based upon

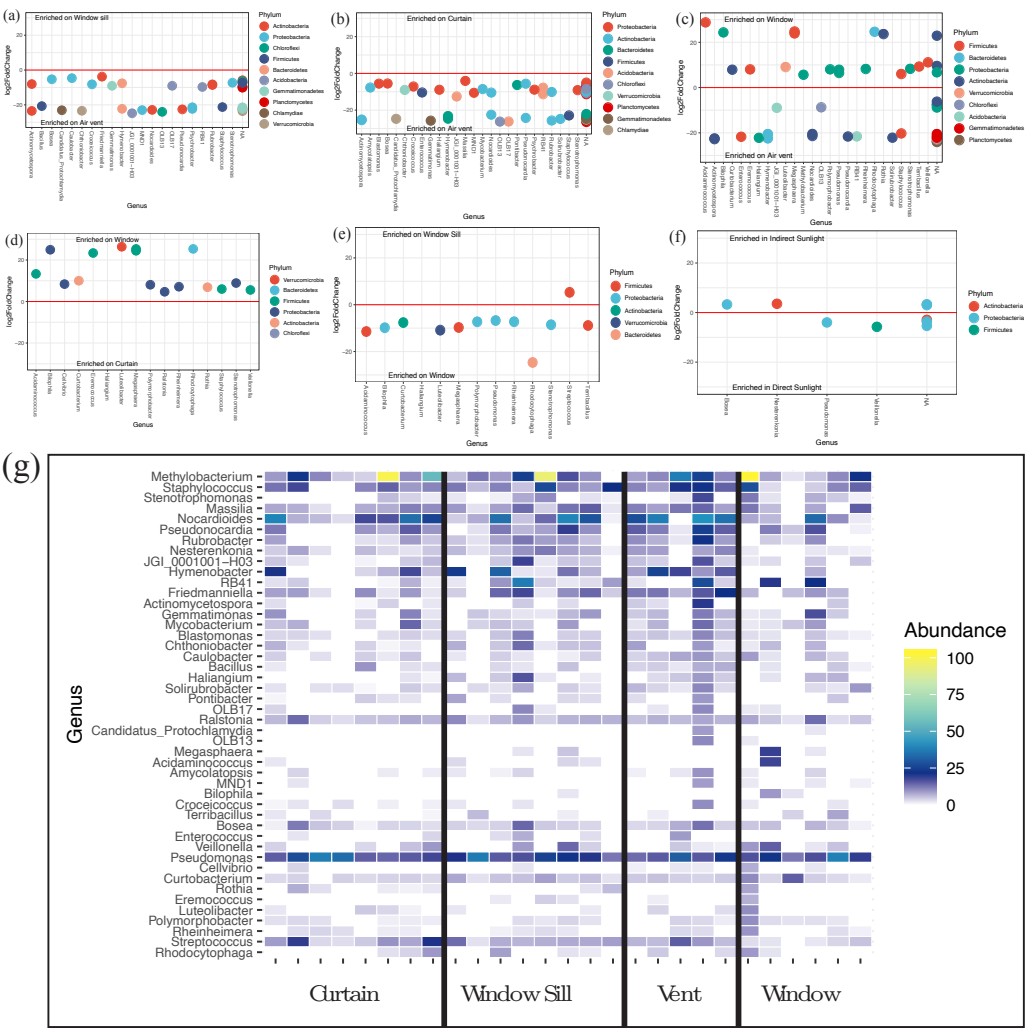

**Figure 5** **Differentially abundant bacteria between sampling locations.** (A–F) Differentially abundant taxa between locations as determined using DESeq2. (G) Heatmap showing relative abundances of viable discriminant taxa. Warmer colors correspond to higher abundances; white tiles indicate those taxa were not detected in particular samples (columns). Columns are individual viable environmental sample communities, where sampling location is indicated by the labels on the *x*-axis.

this small sample set, neither the lighting condition nor the location demonstrates a significant role in the determination of bacterial community composition. However, the failure to capture a larger number of comparable microbial communities from surfaces in direct and indirect sunlight within the same patient room hinders further analysis into the effect of direct vs. indirect sunlight, warranting further investigation.

In the present study, west-facing rooms harbored significantly more bacteria than rooms facing other directions, suggesting that sunlight intensity or exposure time is a significant factor in determining the microbial abundance found within patient rooms. Previous studies demonstrated that microbial communities in model buildings were significantly modified by both ultraviolet and visible spectrum light which mimicked what was found
indoors with standard window glass (*Fahimipour et al., 2018*). All samples were collected between 10 A.M and 11 A.M. PDT. At this time of day, sunlight would have directly entered into rooms, except those facing west. One potential explanation for the observed viable microbial load is that the west-facing rooms had not had sunlight exposure for a longer period of time, and thus viable microbial communities had more time to accrue or recover from the sun exposure of the previous day. This potentially resulted in a higher viable bacteria load during this time of the day compared to the other rooms. However, lacking data on the clinical state of patients, quantification of light exposure in patient rooms, or cleaning records for the period preceding sample collection only permits speculation that a difference in sunlight dosage is responsible for the observed increase in viable bacterial load.

Rooms located at the ends of hallways demonstrated significantly fewer viable bacteria than more centrally located spaces, suggesting that the increased traffic around the central kiosk from nurses, doctors, providers, administrators, and hospital visitors throughout their daily activities contributes to the higher observed bacterial load. Room occupancy on this ward is high, thus rooms further from the centrally-located nurses' station are typically occupied as often as centrally-located rooms in 13K at OHSU. Patients on this ward present with similar medical acuity, and it can be theorized that staff spend similar amounts of time in each room, yet it has been shown that patients further from nursing stations receive less care time and have worse outcomes (*Yi & Seo, 2012*; *Lu et al., 2014*). Thus, it can be assumed that there is more human traffic in the center of the ward, and is likely responsible for the observed gradient of total abundance in rooms along the ward, as healthcare staff both deposit more bacteria during activities and affect microbial transit indoors. Healthcare staff activities contribute to the spread of bacteria throughout hospital wards (*Pittet et al., 1999*; *Perry, Marshall & Jones, 2001*; *March et al., 2010*), and previous work has demonstrated that occupancy (*Meadow et al., 2015*), space-use, and building topology affect indoor microbial communities (*Kembel et al., 2014*).

*Clostridioides difficile* is an anaerobic bacterium that causes extreme dehydration, excessive diarrhea, and colitis (*James, 2013*; *Taylor et al., 2016*), and is among the most common causes of HAIs in the United States (*Carrico et al., 2013*). Viable *C. difficile* was not detected using qPCR in this study; however, as an obligate anaerobe, *C. difficile* is unable to survive outside of its host in a vegetative state for more than a few hours (*Jump, Pultz & Donskey, 2007*). To circumvent this, *C. difficile* forms an endospore, a resilient outer shell, to protect the dormant cell until favorable conditions arise, up to 5 months (*Gerding, Muto & Owens Jr, 2008*). In addition to protection from the outside environment, this spore can also make it difficult to extract DNA, potentially preventing quantification.

MRSA has been linked to bacteremia, endocarditis, and infections of skin and soft tissue infections, bone and joints infections (*Turner et al., 2019*). MRSA was found in only two samples in the present study, and at low abundance. *Staphylococcus aureus (S. aureus)* typically has a high minimum infective dose, owing to its need to form a biofilm for both survival and virulence (*Greig, Todd & Bartleson, 2010*; *Public Health Agency of Canada, 2012*). However, this does not preclude the potential that these small communities of MRSA could potentially increase and pose a risk. MRSA is capable of surviving a variety of

conditions, and can colonize and remain inside an individual without causing symptoms for months at a time (*Public Health Agency of Canada, 2012*; *Turner et al., 2019*). Even more concerning, communities of *S.aureus* have demonstrated the ability to participate in a wide variety of horizontal gene transfer, allowing MRSA of varying pathogenicities and transmission capabilities to develop within the hospital room microbiome by interacting with other bacteria (*Wielders et al., 2001*; *Brody et al., 2008*; *Li et al., 2012*; *Otto, 2013*).

The results of this study are potentially further influenced by unique patient microbiota (*Meadow et al., 2015*) which imprint upon the room, rapid patient turnover which may prevent microbial deposition in rooms (*Meadow et al., 2015*), and the use of harsh cleaning chemicals and different cleaning protocols, all of which presumably alter the total abundance and diversity of the bacterial community within the hospital ecosystem. In particular, bacteria that can survive frequent, harsh cleanings may be better able to withstand variations in environmental conditions within patient rooms (*Velazquez et al., 2019*). While the unique design and occupancy of individual buildings make it difficult to compare indoor microbial communities between structures, it is important to note that many observational studies of microbial communities within the built environment, such as office spaces and homes, identify thousands of unique taxa (*Kembel et al., 2012*; *Stenson et al., 2019*). The present study identified approximately 1000 taxa, despite the high occupancy and activity level typical of a hospital, lending credence to the idea that patient rooms with regular cleaning are a harsh environment for bacteria, and that this cleaning regime is a stronger selection force than sunlight exposure.

## CONCLUSION

This study presents initial data and trends of viable bacterial communities associated with the window-components of patient rooms in a hospital. Viable *Clostridioides difficile* and Methicillin-resistant *Staphylococcus aureus* , two common healthcare-associated opportunistic pathogens, were not detectable and detected at very low abundance, respectively. Patient rooms with west-facing windows demonstrated increased viable microbial load, and viable bacterial abundance was found to be negatively correlated with patient room distance from the central nurses' station. Overall, hospital window components possess similar microbial communities to other previously identified room locations known to act as reservoirs for microbial agents of hospital-associated infections.

## ACKNOWLEDGEMENTS

The authors would like to thank Oregon Health & Science University for help in collecting samples from patient rooms. The authors would like to thank Georgia MacCrone for her help in processing samples. The authors would like to thank the University of Oregon Genomics and Cell Characterization Core Facility (GC3F) for their expertise and resources in Next-Generation sequencing. This work benefited from access to the University of Oregon high performance computer, Talapas.

### Funding

This work was funded by membership dues of View, Inc. to the University of Oregon Institute for Health in the Built Environment Industry Consortium. The funders had no role in study design, data collection and analysis, decision to publish, or preparation of the manuscript.

### Grant Disclosures

The following grant information was disclosed by the authors:
University of Oregon Institute for Health in the Built Environment Industry Consortium.

### Competing Interests

The authors declare there are no competing interests.

### Author Contributions

- Patrick F. Horve performed the experiments, analyzed the data, prepared figures and/or tables, authored or reviewed drafts of the paper, and approved the final draft.
- Leslie G. Dietz performed the experiments, authored or reviewed drafts of the paper, and approved the final draft.
- Suzanne L. Ishaq, Jeff Kline, Mark Fretz and Kevin G. Van Den Wymelenberg conceived and designed the experiments, authored or reviewed drafts of the paper, and approved the final draft.

### DNA Deposition

The following information was supplied regarding the deposition of DNA sequences:
The raw sequencing data from this project are available in the National Center for Biotechnology Information (NCBI) Sequence Read Archive (SRA) BioProject ID: PRJNA610453.

### Data Availability

The code for all analyses and raw abundance data are available at Github: Horve, Patrick (2020). ''Windows-as-Potential-HAI-Reservoirs''. Github. Data and code. https://github.com/BioBE/Windows-as-Potential-HAI-Reservoirs.

### Supplemental Information

Supplemental information for this article can be found online at http://dx.doi.org/10.7717/peerj.9580#supplemental-information.

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
