# Peer review of "Viable bacterial communities on hospital window components in patient rooms"

_PeerJ, doi:10.7717/peerj.9580_

## Round 0.1 · original submission · Minor Revisions

Can you, please, examine the comments of the reviewers and provide a revised version. If you do not agree with some comments, please explain why.

·

Basic reporting

The study is well written, has appropriate references (mostly) and is professionally laid out.

As a preliminary study, the results only provide a brief glimpse to help test the primary hypothesis, and more work will need to be done to provide a compelling test.

Experimental design

The major flaw in this design is the absence of time. The authors admit this. Longitudinal observation would have provided evidence to refute or validate many of the potential claims in this paper. In the absence of time series any interpretations are speculative at best.

Validity of the findings

Overall the study is a basic snapshot of the hospital environment, but without time or environmental analysis, these data have immediate limited value as none of the speculative interpretations can be tested with the data as presented here.

Additional comments

Ln 245-248: Fig 2C – the abundance profiles are incredibly slight, what is the median, mean and SD for each of those? What is the p?

Ln 270-284 – as this is proportional data, refer to it as such, to differentiate from the qPCR abundance data.

Fig. 4b – I am again surprised that the alpha diversity is significantly different on the window? The distribution of scores is definitely tighter, as such maybe using a statistic that takes into consideration spread would be valuable?

Surprised not to see any environmental data – light, relative humidity, temperature, that could have been used to interpret trends? i.e. were rooms closer to the nursing station environmentally different?

Statements like “Further, this study confirmed a spatial pattern to bacterial abundance based on putative occupancy density, which implies that human occupants may have a stronger effect on indoor microbial communities than ambient light exposure.” – should be tempered, as proximity to the nursing station may have no impact at all on bacterial burden per se, let alone occupant density – without occupancy detection being measured, this is just one possible explanation? Another might be different disease state, or patient age, or equipment?

The discussion is very long for such a limited results paper, there is a certain tendency towards over interpretation. For example, referencing Hirai 1991 regarding nutrient retention on glass, and then using that as a potential explanation is another example of one potential explanation, but this is not substantiated by the reference.

Remember, you have 1 single time point, so talking about glass as a barrier for bacterial development is hyperbolic – viable cells doesn’t mean they are active, growing or persisting. They may have just landed there?

There is no evidence to support bacterial adhesion to the glass – “This suggests that bacteria able to successfully adhere to the glass are able to survive intact.” Static interaction between positively charged dust particles could also result in temporary adhesion of dust, which may contain bacterial cells.

“and a lack of regular cleaning and frequent reinoculation from patients and visitors in the room might explain the higher differential abundance of these specific bacteria on the glass.” – the differences were slight at best, without a temporal analysis this is extreme speculation.

·

Basic reporting

Horve et al estimate bacterial load and community composition associated with 6 different sites on window components in hospital rooms. The paper is largely descriptive, but the authors find greater numbers of viable bacteria in widows that have not recently been exposed to direct sunlight, and in rooms which are more central in the hospital floor plan.
The paper is well written, the figures are easy to interpret, and the authors make reasonable and well referenced inferences from their data.

Experimental design

All samples were taken during an hour long sampling window on a single day, which necessarily limits the authors' ability to test their hypothesis that recent direct sunlight lowers viable bacterial cell counts (i.e. it would have been nice to have measurements from west facing windows taken at sunset). Still, the authors do not over extrapolate from their data, and provide meaningful context to their hypotheses. I have no personal expertise in DNA extraction/amplification, but their methodology is well described and appears rigorous. The statistical analyses are reasonable and well described. The methods are sufficiently descriptive that the study could be easily replicated.

Validity of the findings

All underlying data has been provided. The conclusions are well stated and speculation is identified as such and supported by citations to other research.

Additional comments

Two minor points in the abstract:

Lines 32 & 87: I assume the authors mean ‘does not raise cause for alarm’

Lines 43-44: This seems like an incomplete sentence, perhaps the authors mean to say something like ‘and community composition was assessed using Illumina…’

---

## Round 0.2 · accepted · Accept

Thank you for correcting your paper as per the reviewers' suggestions. I guess this has now resulted in a stronger paper.